# JETFORMER: AN AUTOREGRESSIVE GENERATIVE MODEL OF RAW IMAGES AND TEXT

**Michael Tschannen,**[*] **André Susano Pinto,**[*] **Alexander Kolesnikov**[*]
Google DeepMind
{tschannen,andresp,akolesnikov}@google.com

## ABSTRACT

Removing modeling constraints and unifying architectures across domains has been a key driver of the recent progress in training large multimodal models. However, most of these models still rely on many separately trained components such as modality-specific encoders and decoders. In this work, we further stream-line joint generative modeling of images and text. We propose an autoregressive decoder-only transformer—JetFormer—which is trained to directly maximize the likelihood of raw data, without relying on any separately pretrained components, and can understand and generate both text and images. Specifically, we leverage a normalizing flow model to obtain a soft-token image representation that is jointly trained with an autoregressive multimodal transformer. The normalizing flow model serves as both an image encoder for perception tasks and an image decoder for image generation tasks during inference. JetFormer achieves text-to-image generation quality competitive with recent VQVAE- and VAE-based baselines. These baselines rely on pretrained image autoencoders, which are trained with a complex mixture of losses, including perceptual ones. At the same time, JetFormer demonstrates robust image understanding capabilities. To the best of our knowledge, JetFormer is the first model that is capable of generating high-fidelity images and producing strong log-likelihood bounds.

## 1 INTRODUCTION

The "Bitter lesson" (Sutton, 2019) has been the prime force behind the recent progress in machine learning and artificial intelligence research. It suggests that general-purpose methods which effectively leverage large amounts of compute and data prevail over specialized techniques designed by domain experts. Arguably the most prominent examples in this context are transformer decoder-only models trained for next-token prediction (Vaswani et al., 2017; Radford et al., 2018), that outperform task-specific NLP systems, and transformer encoders in computer vision (Dosovitskiy et al., 2021; Strudel et al., 2021; Li et al., 2022), that achieve better quality than CNN-based models.

This trend is also visible in the current pursuit of extending LLMs to both understand and generate multiple modalities such as text and images with a single model. A powerful paradigm in the literature (Aghajanyan et al., 2022; Kim et al., 2023; Aghajanyan et al., 2023; You et al., 2023) is to model the image tokens using discrete tokens obtained via (VQ)VAEs (van den Oord et al., 2017; Esser et al., 2020; Ramesh et al., 2021). One limitation of these approaches is that the conversion from image into tokens and vice-versa is performed by a separate, frozen, modality-specific and lossy encoder (and decoder) trained ahead of time. As a result, this image encoder may be agnostic to the actual task at hand and limit the performance of the resulting model (Dong et al., 2023; Pan et al., 2024; Xu et al., 2024).

To obtain a general architecture that can generate multiple modalities but does not have (limiting) components pretrained ahead of time, we develop a new generative model: the JetFormer. It can be trained from scratch and optimized end-to-end for the log-likelihood of raw training data. We demonstrate this for text and pixels. To this end, we combine a normalizing flow (Dinh et al., 2016; Kingma & Dhariwal, 2018) for computing a soft-token image representation with a decoder-only transformer (Vaswani et al., 2017) and a soft-token Gaussian mixture loss (Tschannen et al., 2024).

---

[*]Equal contribution.

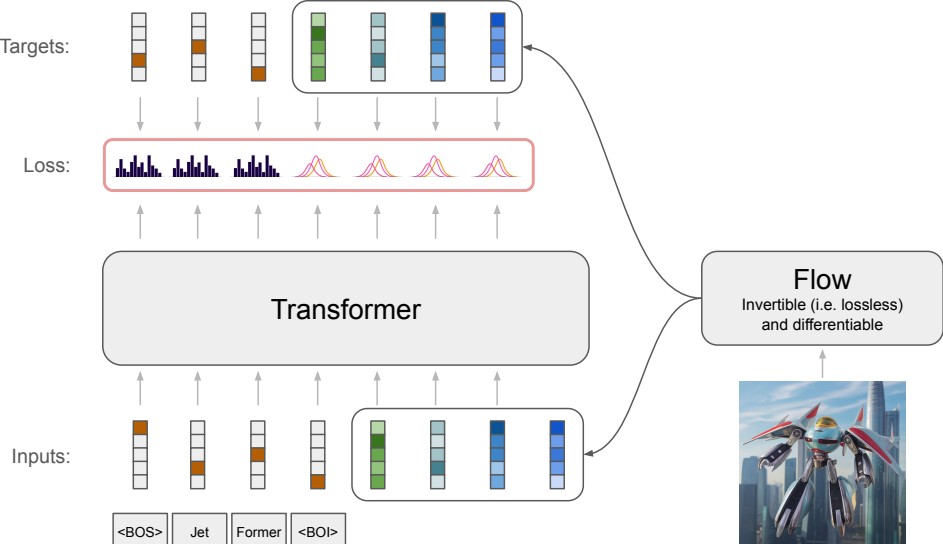

Figure 1: Visualization of JetFormer training with teacher-forcing. The ground truth consists of text tokens and images. The images are converted into soft tokens via a normalizing flow. The sequence of tokens is processed by an auto-regressive transformer and its outputs parameterize either a discrete distribution or a Gaussian mixture depending whether the target is a discrete or a soft token.

The key insight behind the JetFormer model, is that a powerful normalizing flow (which we call a "jet", hence the model name) can be used to encode images into a latent representation suitable for autoregressive modeling. Intuitively, raw image patches encoded as pixels have very complex structure, which makes direct autoregression futile: to date, there is no convincing demonstration of this. At the same time, the flow model is lossless and can be trained together with the (multimodal) autoregressive model end-to-end. At inference time an image decoder is readily available, since our flow model is invertible in closed form.

Although we only optimize for log-likelihood, it is worth noting that doing so naively does not guarantee being able to generate images with global coherence. Similarly to the vast majority of work on high-fidelity image generation (Esser et al., 2020; Dhariwal & Nichol, 2021; Ho & Salimans, 2022), we guide the model to focus on the high-level information. To this end, we explore two approaches.

First, we introduce a novel technique that is based on image augmentation during training. The main idea is to add Gaussian noise during training, but reduce it through the course of training. Intuitively, this pushes the model towards prioritizing high-level information early on; see Section 3.2.1 for more details. Even though a noise curriculum during training is inspired by diffusion models (Sohl-Dickstein et al., 2015), it is very different on the technical level, and the resulting model does not perform progressive image denoising at inference time.

Second, we explore two ways of managing redundancy in natural images. JetFormer readily allows excluding a subset of redundant dimensions from the autoregressive model. As an alternative, we explore PCA for reducing image dimensionality.

We conduct experiments on ImageNet class-conditional image generation and on web-scale multimodal generation, thereby demonstrating the JetFormer works and scales to both text-to-image generation and vision-language understanding with a single model.

In summary, our contributions are:

- We present JetFormer, a generative model composed of a transformer and a normalizing flow that can be trained from scratch to model text and raw pixels jointly, end-to-end.

- We show that image augmentation based on a noise curriculum can significantly boost image generation quality of such likelihood-based models.

- We demonstrate our proposed end-to-end model is competitive with less flexible techniques when trained on web-scale data, and can generate both images and text.

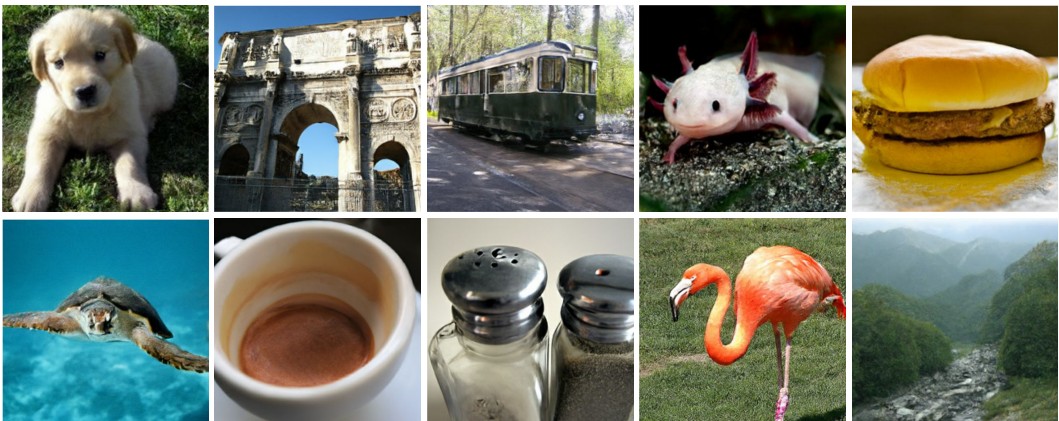

Figure 2: Selected class-conditional $256{\times}256$ samples from the JetFormer-L trained on Imagenet. The samples are produced with the CFG sampler, where CFG strength is equal to $4.0$.

## 2 RELATED WORK

Generating natural images autoregressively as a sequence of discrete-valued (sub-)pixels was extensively explored in the literature using CNNs (Van den Oord et al., 2016b;a; Salimans et al., 2016) or transformers (Parmar et al., 2018; Chen et al., 2020). While achieving excellent results in log-likelihood, these models are computationally expensive and do not scale well to high image resolutions. A related family of models are normalizing flows (Dinh et al., 2014; 2016; Kingma & Dhariwal, 2018; Ho et al., 2019), invertible models which are trained to map image pixels to a simple prior by maximizing log-likelihood. These models scale better but achieve lower likelihood than autoregressive models and empirically fail to generate high-fidelity images, even for low resolutions.

More recently, compressing the high-dimensional image pixel space to a lower-dimensional sequence of discrete tokens via a pretrained, frozen VQ-VAE (van den Oord et al., 2017; Razavi et al., 2019), and then modeling the compressed sequence with a transformer decoder has emerged as a scalable technique for high-fidelity image generation (Esser et al., 2020; Yu et al., 2022a; Ramesh et al., 2021; Yu et al., 2022c; Gafni et al., 2022). To enable semantic compression, VQ-VAEs typically rely on perceptual and GAN losses. Moreover, VQ-VAE-based representations are common in the context of dense prediction tasks (Kolesnikov et al., 2022; Lu et al., 2022; Mizrahi et al., 2024; Mentzer et al., 2024), in particular when modeling multiple modalities jointly. GIVT (Tschannen et al., 2024) showed that combining an autoencoder and an autoregressive transformer can be applied to continuous-valued sequences, by directly modeling feature vectors in the latent space of a VAE, without any quantization. Somewhat related, (Weiss et al., 2021; Nachmani et al., 2023; Meng et al., 2024) explored soft tokens for speech synthesis.

VQ-VAEs are also becoming popular in the context of Vision-Language Models (VLMs). Such models are typically either trained from scratch (Radford et al., 2021; Jia et al., 2021; Wang et al., 2022b; Yu et al., 2022b) on web-scale data or constructed by combining and tuning a pretrained vision encoder and a pretrained language model (Alayrac et al., 2022; Chen et al., 2023b; Wang et al., 2022a; Li et al., 2023; Huang et al., 2023; Liu et al., 2024; Beyer et al., 2024), and can solve a broad range of task which can be cast as text output. To enable pixel outputs for such models a simple way is to extend the text vocabulary with VQ-VAE tokens (Aghajanyan et al., 2022; Kim et al., 2023; Aghajanyan et al., 2023; You et al., 2023; Pan et al., 2024). Other works (Dong et al., 2023; Ge et al., 2024; Zhou et al., 2024) combine VLMs with (latent) diffusion models (Sohl-Dickstein et al., 2015; Dhariwal & Nichol, 2021; Rombach et al., 2022; Saharia et al., 2022; Ramesh et al., 2022) to enable image generation capabilities. JetFormer is related to this category of models, but unlike previous models does not rely on any pretrained (VQ-)VAE vision encoder/decoder.

## 3 METHOD

Modeling natural images with autoregressive transformers poses many obstacles. Doing so in pixel space is a viable approach (Parmar et al., 2018; Chen et al., 2020), but it quickly becomes computationally prohibitive. Even an image of size $256{\times}256{\times}3$ would require predicting/sampling

nearly 200 000 tokens. Alternatively, modeling images at the patch-level to tame the computational complexity creates its own challenges. Each patch is a sample from a complex distribution, and producing all of its dimensions in a single forward pass fails to model pixel interactions.

Currently, the most common approach to overcome these issues is to leverage a standalone image encoder/tokenizer (and decoder/detokenizer) model, which encodes an image as a sequence of (usually discrete) tokens. Such an image encoder performs semantic compression and thus reduces the computational burden. However, this type of approach has significant shortcomings: one needs to tolerate precision loss due to compression and to commit to the image encoder that was trained ahead of time and may not be suitable for the modeling task at hand.

In this paper we overcome both of these issues and propose JetFormer, a generative autoregressive decoder-only model able to model both text and images, and directly learn from the raw training data. The JetFormer model is trained end-to-end, without relying on any lossy modality-specific encoders/tokenizers. Our model is built on two key insights.

First, we use continuous (also called "soft") tokens to represent images. As shown in GIVT (Tschannen et al., 2024), transformer decoders can generate the soft-token-sequence produced by a VAE encoder for high-fidelity image generation. Specifically, GIVT replaces the categorical prediction head with a Gaussian Mixture Model (GMM) that models log-likelihood of soft image embeddings.

Second, instead of a VAE (that needs to be trained ahead of time), we use a normalizing flow model to learn a soft-token image representation suitable for autoregressive modeling. As flow models are lossless by design, they won't suffer from representation collapse and can be trained simultaneously with the transformer model without auxiliary losses, eliminating the necessity to use pretrained encoders and enabling full end-to-end learning of autoregressive transformers from raw images.

Naturally, the above two insights can be combined with the standard transformer for text, forming a simple and unified multimodal model, capable of learning from image and text data.

### 3.1 MODELING IMAGES IN PIXEL SPACE WITH SOFT TOKENS AND NORMALIZING FLOWS

As outlined above, we model an image $x$ using a normalizing flow model (Dinh et al., 2014; 2016; Kingma & Dhariwal, 2018) $f(x)$ that losslessly maps an image into a sequence of embeddings $\{z_1, \ldots, z_n\}$, which we also call "soft tokens". Note that the flow preserves the total number of the input dimensions. These embeddings are then modeled by the deep autoregressive model $p$, where the outputs are modeled with a GMM, as proposed in GIVT. We then maximize the image log-likelihood lower bound $L$:

$$L(x) = \log p(f(x)) + \log \left| \det \left( \frac{\partial f(x)}{\partial x^T} \right) \right|, \quad \text{where} \tag{1}$$

$$f(x) = [z_1, z_2, \ldots, z_n] \quad \text{and} \quad p(z) = \prod_{i=1}^{n} p(z_i | z_{i-1}, \ldots, z_1). \tag{2}$$

Note the log-determinant term arises from the normalizing flow model, as a part of the data log-likelihood, see Dinh et al. (2016). Further, to ensure correctness, we apply image dequantization, as outlined in Theis et al. (2015). This amounts to adding uniform noise $u$ to the input images $I$, s.t. $x = I + u$, where $u \sim U[0, 1]$. This guarantees that we optimize a lower bound on the discrete image log-likelihood. For clarity, we point out that both $p$ and $f$ both have learnable parameters which are optimized with gradient-based method (training via teacher forcing is illustrated in Figure 1).

In simple words, JetFormer for images is an autoregressive model, where inputs and targets are produced by the flow model, which re-encodes input images. Due to the end-to-end nature of the objective, the flow model is incentivized to learn a sequence of embeddings, that makes autoregressive modeling as effective as possible. During inference, the autoregressive model generates a sequence of soft tokens, which then need to be decoded into an image using the inverse of the flow.

### 3.2 IMPROVING MODEL QUALITY FOR NATURAL IMAGES

While JetFormer works out of the box, we have found several modeling enhancements which greatly improve the quality of the generated images. In particular, factoring out latent dimensions, the use of classifier-free-guidance during sampling, and a novel noise curriculum.

**Factoring out redundant dimensions** Natural images are redundant, intrinsically low-dimensional signals with low-frequency components dominating the spectrum. The design of Jet-Former enables a simple and effective way to leverage this observation and improve model quality, while also reducing computational burden.

The key observations is that not all output dimensions of the invertible flow need to be further processed by the autoregressive model. We can model a subset of dimensions (i.e. a subset of channels) with a Gaussian distribution, $p_\mathcal{N}$, and the remaining ones with the autoregressive transformer:

$$L(x) = \log p(\hat{z}) + \log p_\mathcal{N}(\tilde{z}) + \log \left| \det \left( \frac{\partial f(x)}{\partial x^T} \right) \right|, \text{ where } [\hat{z}, \tilde{z}] = f(x)$$

Intuitively, we expect redundant dimensions to be "factored out" as $\tilde{z}$, as they do not require further heavy processing. We verify our intuition empirically in the experimental section and in Figure 6c.

As a strong baseline to the above approach, we also consider a more direct approach to handle redundancy in images. Before feeding $x$ to the flow model, we reshape it into a sequence of flattened patches and apply a learnable, invertible linear map $W$ along the channel dimension. We want this map to learn to separate important dimensions of the flattened patches from redundant ones. To this end, we feed the first $d$ channels of its output $xW^\top$ to the normalizing flow and model the remaining channels with a Gaussian distribution. Intuitively, given the simplicity of the transform $W$ applied before factoring out part of the sequence, minimizing the NLL while training will ensure that the hard-to-model part of the sequence is modelled by JetFormer, whereas the low-level noise will be mapped to the Gaussian prior. This is similar to the reasoning behind probabilistic PCA (Tipping & Bishop, 1999). Indeed, we observe that the transform learned by the model is close to applying PCA to image patches (see Figure 6d), and we observe that when initializing $W$ with PCA and freezing it we obtain similar results. See Appendix B for formal definition of this approach.

**Classifier-free guidance** Following common practice in the diffusion and autoregressive image modeling literature, we employ classifier-free guidance (CFG) (Ho & Salimans, 2022) which was previously shown to substantially improve sample quality. We reuse the distribution-based variant implemented via rejection sampling for GMMs from (Tschannen et al., 2024) without modification.

### 3.2.1 RGB Noise curriculum during training

It is common to explicitly factorize data into semantically meaningful parts to improve image quality. One approach is to model RGB pixels as sequences of increasing color depth and/or resolution (Kolesnikov & Lampert, 2017; Menick & Kalchbrenner, 2019; Nash et al., 2021). Similarly, adding noise to RGB pixels is closely related to reducing the color depth and effective resolution. This has led to the interpretation of diffusion models, where denoisers are trained at different noise levels according to a predefined noise schedule, as learning a hierarchical representation in pixel space induced by the noise schedule (Kingma & Gao, 2023; Dieleman, 2024).

Building on this intuition, we alter the training procedure by introducing a "noise curriculum": additive Gaussian pixel noise during JetFormer training. The noise is strongest in the beginning of the training and decays towards zero. We use cosine decay schedule for the noise standard deviation.

In the beginning of the training, when strong (high-variance) noise is added to the image, JetFormer learns to model coarse image information (see Figure 6b). As training progresses, the model gradually learns a finer level of detail, while "remembering" previously learned patterns. In the end of the training, JetFormer uses the correct distribution for training. Intuitively, this scheme prioritizes modeling of high-level image structure without sacrificing overall performance.

Importantly, unlike in diffusion, the noise curriculum merely acts as a data augmentation during training. The model is not conditioned on the noise magnitude and, at inference, an image is not gradually denoised, but generated autoregressively in the latent space of a normalizing flow.

For an integer-valued RGB image $I$, the noisy image is obtained as $\lfloor I + \sigma_t N(0, \mathrm{I}) \rfloor$, where the noise scale $\sigma_t$ as a function of the training progresses $t \in [0, 1]$ follows the cosine schedule $\sigma_t = \sigma_0 \frac{1 + \cos(t\pi)}{2}$. The shape of the noise schedule is visualized in Figure 3.

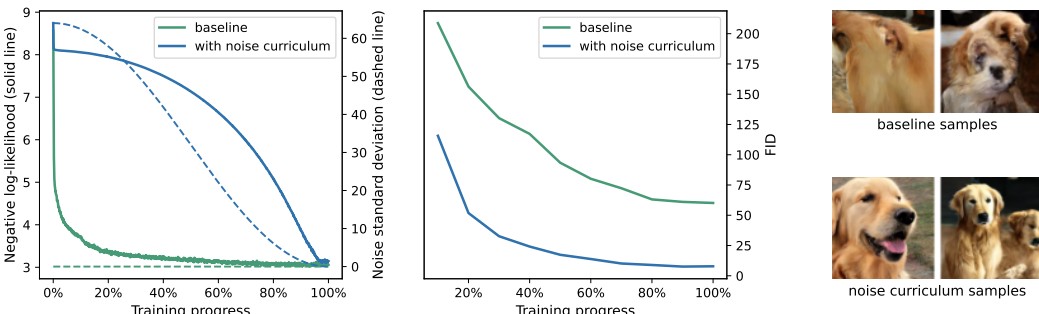

Figure 3: Training with noise curriculum on RGB input images. The left plot demonstrates negative log-likelihood progression (solid lines) and the noise strength schedule (dashed lines). Importantly, both NLL curves land in a very similar points, despite visual quality measured by FID (middle plot) and the typical samples (right plot) being very different: we observe that noise curriculum guides the JetFormer towards modeling high-level image structure.

### 3.3 JOINT GENERATIVE MODELING OF PIXELS AND TEXT

We explore JetFormer for multimodal generative modeling, where the model can perform both discriminative and generative tasks on all modalities, focusing on images and text. Sophisticated models of this class are trained on interleaved sequences of images and text (Aghajanyan et al., 2022; Kim et al., 2023; Aghajanyan et al., 2023), often with post-training refinement, which enables few-shot image-to-text (e.g. captioning) and text-to-image (e.g. image editing) capabilities. Here, we follow (Kim et al., 2023; You et al., 2023) and consider image-text pairs from the web as a proxy for more complex, interleaved setups, and do not involve a post-training step. While conceptually simple, this allows us to explore vision-language understanding tasks such as captioning and VQA, as well as text-to-image generation. Extending the image generation approach discussed in the previous section to this setup is straight-forward: We simply extend the transformer backbone generating soft tokens to modeling language tokens produced by a standard language tokenizer with a separate prediction head and a softmax.

We train on sequences of both image tokens followed by text tokens, and vice versa, only applying a loss to the second part (modality) of the sequence. We use the respective negative log-likelihood, and a weight to balance the two. We observed that applying the loss to the full sequence leads to worse results, possibly because predicting an image prefix effectively means unconditionally modeling web images, which is very challenging. We expect this to change for interleaved sequences, which may provide a stronger conditioning signal.

For text-to-image generation, the text prefix acts as a conditioning, and image generation is performed as described in Section 3.1. For image-to-text generation the normalizing flow acts as an image encoder. The model uses the same soft token space for generation and understanding.

## 4 EXPERIMENTS

**Architecture** We rely on the simple, transformer-based design from (Kolesnikov et al., 2024) for the normalizing flow model. This design uses affine coupling layers (predicting an element-wise scale and shift) consisting of spatial and channel-wise splitting functions to split the activations in two parts, and a stack ViT blocks (Dosovitskiy et al., 2021) applied to half of the activations to infer the affine transform. Here, we only use channel-wise splitting as we found in preliminary experiments that spatial splitting did not improve modeling quality. We set the depth to 32 coupling blocks, each consisting of a stack of 4 or 6 ViT blocks of width 512, and with 8 attention heads. The (input and output) feature shape for this model is $(\frac{H}{p}, \frac{W}{p}, 3p^2)$ when feeding the full image, and $(\frac{H}{p}, \frac{W}{p}, d)$ when using a dense invertible map or PCA to reduce dimensionality prior to applying flow. For image size $H = W = 256$ and patch size $p = 16$ this amounts to $256 \times 768$, after flattening spatial dimensions and with dimensionality reduction to $d = 128$ to $256 \times 128$.

Table 1: Comparison of JetFormer trained for 500 epochs on ImageNet256 with baselines from the literature. For large enough scale JetFormer approaches models without components pretrained in an extra step.

|  | extra step | FID | Precision | Recall | NLL |
|---|---|---|---|---|---|
| BigGAN-deep (Brock et al., 2018) | – | 6.95 | 0.87 | 0.28 | |
| ADM-G (Dhariwal & Nichol, 2021) | – | 4.59 | 0.82 | 0.52 | |
| LDM-4-G (Rombach et al., 2022) | VAE | 3.60 | 0.87 | 0.48 | |
| VQGAN (Esser et al., 2020) | VQ-VAE | 5.20 | | | |
| ViT-VQGAN (Yu et al., 2022a) | VQ-VAE | 3.04 | | | |
| GIVT-Causal (Tschannen et al., 2024) | VAE | 3.35 | 0.84 | 0.53 | |
| JetFormer-B | – | 7.25 | 0.72 | 0.44 | 3.06 |
| JetFormer-L | – | 6.64 | 0.69 | 0.56 | 3.05 |

For the latent autoregressive decoder-only backbone, we rely on the Gemma architecture (Gemma Team et al., 2024). We consider 3 different model shapes amounting to 350M, 750M and 1.3B parameters which are largely inspired by previous decoder-only models for image generation (Esser et al., 2020) (see Appendix A). For the GMM prediction head, unless explicitly noted, we set the number of mixtures $k = 1024$, use multivariate Gaussians of dimension $d = 128$ with diagonal covariance. For text we use a sentencepiece tokenizer with vocabulary size 32k trained on the English portion of the C4 corpus made available by Raffel et al. (2020). We set the maximum number of text tokens to 64, but do not explicitly model padding tokens (i.e. during training we mask out attention elements corresponding to padding tokens and adapt the RoPE positions to skip them). When training for class-conditional image generation, we use a prefix of 16 learned tokens per class (instead of a single one), as we observed that this improves sample quality.

**Training recipe** We train the latent decoder-only backbone on concatenated sequences of discrete text and soft tokens (flow latents) via teacher forcing for NLL according to the respective distribution of the tokens (categorical for text tokens and GMM for soft tokens). Whether the sequence is an image-text or text-image sequence is sampled at random for every example. Normalizing flow (or learnable invertible patch embedding) is trained along with the transformer-backbone end-to-end. This does not require any dedicated techniques thanks to the NLL for the GMM being differentiable both w.r.t. to the GMM parameters as well as the soft tokens/features it is evaluated w.r.t. When training for captioning with an image prefix (i.e. for image-text sequences, where normalizing flow serves as a vision encoder), we stop the gradient at the flow output during pretraining. We did not observe improved performance when passing the gradients.

We use the Adam optimizer with learning rate $10^{-3}$, decoupled weight decay of $10^{-4}$, $\beta_2$ parameter 0.95 and clip the gradient norms to 1.0. We set the batch size to 4k. We also apply dropout with probability 0.1 at the output of the self-attention and MLP blocks, which we found to improve image sample quality. For both class conditional image generation and text-to-image generation we drop the conditioning with 10% probablilty and replace it with a learned `[NOLABEL]` token for CFG. Unless explicitly stated, we apply the RGB noise schedule to the input images described in Section 3.1 with initial noise standard deviation $\sigma_0 = 64$ (for pixel values in the range $[0, 255]$). We decay to 0 for ImageNet setup and to 3 for multimodal setup. Inspired by the VAE-based GIVT, where for every example a latent is sampled from the VAE encoder (i.e. approximate posterior) which might have a regularizing effect, we add Gaussian noise with standard deviation 0.3 to the flow latents. We normalize the image NLL to bits per dimension as is common in the image generative modeling literature and apply a weight of 0.0025 to the text NLL such that the loss magnitude per token is roughly identical for both modalities.

**Training data** For training class-conditional image generation models we use ImageNet1k (Russakovsky et al., 2015). For multimodal generation we rely on the image-text pairs from WebLI data set (Chen et al., 2023b). In both cases, we resize the images so that the shorter side is 256 pixels while preserving the aspect ratio and extract a $256{\times}256$ central crop. Besides the RGB noise described earlier we do not apply any augmentation, except for random left-right flipping on Im-

Table 2: Summary of the main JetFormer (trained for 100 epochs) ablations performed on class-conditional ImageNet256 generation. We demonstrate relative importance of various components and highlight the interesting interplay between PCA preprocessing and the noise curriculum.

|  | FID | Precision | Recall | NLL |
|---|---|---|---|---|
| JetFormer-B | 7.84 | 0.75 | 0.39 | 3.14 |
|    no normalizing flow | 117.76 | 0.17 | 0.32 | 6.84 |
|    no noise curriculum | 44.71 | 0.45 | 0.28 | 3.05 |
|    no factored-out dimensions | 17.29 | 0.65 | 0.26 | 3.13 |
|    no end-to-end training | 11.16 | 0.68 | 0.33 | 3.08 |
|    learned inv. projection | 10.19 | 0.73 | 0.32 | 4.78 |
|    no GMM (Gaussian loss only) | 9.46 | 0.77 | 0.30 | 3.14 |
|    single class token | 8.85 | 0.73 | 0.37 | 3.14 |
| PCA preproc. + JetFormer-B | 8.79 | 0.77 | 0.35 | – |
| PCA preproc. + JetFormer-B (no noise cur.) | 13.16 | 0.71 | 0.31 | – |

ageNet1k. On ImageNet1k, we train for 100 epochs in ablations, and 500 epochs otherwise. On WebLI, we train for 1B examples seen per modality, so 1B examples for text-to-image only models, and 2B total for models that also support image-to-text (understanding) tasks.

**Evaluations and metrics** Following the diffusion literature, we use the ADM FID evaluation suite (Dhariwal & Nichol, 2021) (with 50k reference samples) and precision/recall (Sajjadi et al., 2018) to assess image sample quality on ImageNet1k. For text-to-image, we adopt the common MS-COCO FID-30k, generating images for captions from 30k randomly sampled COCO validation images and evaluating FID against reference statistics from the full COCO validation set. We report this metric both zero-shot and after fine-tuning on the COCO training set. As image understanding tasks we consider ImageNet zero-shot classification and fine-tune for image captioning (reporting CIDEr score) and visual question answering (VQA, measuring VQAv2 accuracy).

## 4.1 CLASS-CONDITIONAL IMAGE GENERATION

Table 1 shows the sample quality of JetFormer trained on ImageNet with image resolution of $256 \times 256$ along with some baselines from the literature. Model samples are show in Figure 2 and Appendix F. Despite being an explicit NLL model and not using advanced image encoders, our Jet-Former model is competitive with those baselines. Interestingly, JetFormer has high recall of $0.56$. We hypothesize that this is a consequence of our model being an explicit log-likelihood model and thus not suffering from the mode collapse.

Table 2 shows ablations of key design choices of JetFormer:

- Removing the normalizing flow results in a catastrophic loss in performance. This confirms our intuition that re-encoding image pixels with a flow model is essential.
- Omitting the noise curriculum further results in much worse results.
- We also confirm that not factoring out dimensions after the flow model leads to quality loss.
- First training the normalizing flow to minimize NLL w.r.t. a Gaussian prior, and then training the autoregressive transformer on the latent representation of the frozen flow model (factoring out redundant dimensions of the frozen latent space with a learnable, invertible linear map) results in lower sample quality than end-to-end training.
- Factoring out dimensions with a learnable linear mapping before the flow (as described in Sec. 3.2) leads to better results, but underperforms post-flow factoring out.
- Reducing the number of GMM components from 1024 to 1 for the soft-token loss result in a relatively mild performance drop and a significant drop in recall, indicating more mixtures enable a better coverage of the distribution.
- Finally, doing class-conditioning with a single class token (instead of 16 tokens) leads to slight performance drop, likely due to the weaker conditioning signal.

Interestingly, we observe that modeling images after the PCA transform leads to somewhat worse results. However, in the presence of PCA, the noise curriculum becomes less important. It highlights

Table 3: Comparison with baselines from the literature for text-to-image generation (0-shot COCO FID-30k, $256\times256$). We included autoregressive (first group) and diffusion models (second group) which use raw image-text data (without e.g. re-captioning) and do not rely on pretrained text encoders. Models with image understanding capabilities are marked with T&I. FID (ft.) is obtained when fine-tuning on the COCO training set, and NLL is measured on the COCO validation set after fine-tuning. JetFormer is the only method which does not rely on any extra step.

| | extra step | #param. | FID | FID (ft.) | NLL (ft.) |
|---|---|---|---|---|---|
| DALL-E (Ramesh et al., 2021) | VQ-VAE | 12B | 27.50 | | |
| CogView (Ding et al., 2021) | VQ-VAE | 4B | 27.10 | | |
| CogView2 (Ding et al., 2022) | VQ-VAE | 6B | 24.00 | 17.50 | |
| ARGVLT (T&I) (Kim et al., 2023) | VQ-VAE | 0.45B | 16.93 | | |
| MAGVLT (T&I) (Kim et al., 2023) | VQ-VAE | 0.45B | 12.08 | | |
| Make-A-Scene (Gafni et al., 2022) | VQ-VAE | 4B | 11.84 | 7.55 | |
| LDM-KL-8-G (Rombach et al., 2022) | VAE | 1.45B | 12.63 | | |
| GLIDE (Nichol et al., 2022) | Super-res. | 6B | 12.24 | | |
| DALL-E-2 (Ramesh et al., 2022) | Super-res. | 5.2B | 10.39 | | |
| JetFormer-L (T&I) | – | 2.75B | 20.86 | 13.70 | 3.86 |
| JetFormer-L | – | 2.75B | 18.63 | 13.07 | 3.85 |

the importance of prioritizing the high-level information in images, if visual quality is the end goal. It also shows that manual preprocessing may reduce the necessity for various modeling tricks, yet it also shows that full end-to-end modeling prevails when done right.

## 4.2 EFFECT OF THE NOISE CURRICULUM

We have experimented with different levels of initial noise, and generally find that for $256\times256$ images the initial noise with standard deviation of $64$ annealed to $0$ during the training to be the best for the ImageNet setup. The initial noise levels of $128$, $64$, $32$ and $0$ result in FID $8.62$, $7.84$, $8.59$ and $44.71$ respectively. For the multimodal setup we observe that annealing the noise standard deviation to $3$ (thus leaving a small level of noise) improves FID. More analysis of the interplay between NLL and FID is presented in Appendix C.

In Figure 3 we demonstrate the effect of noise curriculum with the initial standard deviation of $64$. First, we observe that, in comparison to the noise-free baseline, the final NLL is not affected dramatically. However, as would be expected, initial stages of training have much worse NLLs due to strong noise. Second, we demonstrate that FID drastically improves with the noise curriculum being enabled. This is also evident by the final samples from the noise-free model and model with noise. The latter has much more pronounced emphasis on the high-level image structure.

## 4.3 MULTIMODAL GENERATION AND UNDERSTANDING

Figure 4 shows multimodal JetFormer models of different sizes, trained for text-to-image generation only (T2I) and both T2I and image-to-text generation (T&I). Zero-shot samples are show in Appendix E. In both cases, increasing the model size improves quality. Training models for mixed T&I generation leads to a reduction in T2I generation quality, but also equips the model with vision-language understanding capabilities.

Tables 3 and 4 compare T2I and T&I JetFormer models with models from the literature. Similar prior work trained in generative fashion on image-text pairs relying on pretrained VQ-VAE (ARGVLT, MAGVLT (Kim et al., 2023)) achieves better performance on T2I generation, but lags in understanding performance, which highlights the benefits of having access to unquantized end-to-end learned features for understanding. JetFormer also approaches the understanding performance of image-text models pretrained for understanding such as CLIP and CapPa, although these models have a significantly smaller size.

Finally, we present a T&I baseline using a pretrained, frozen VAE instead of an end-to-end trained normlizing flow in Appendix C and find that JetFormer clearly outperforms this baseline in T2I generation and all I2T tasks.

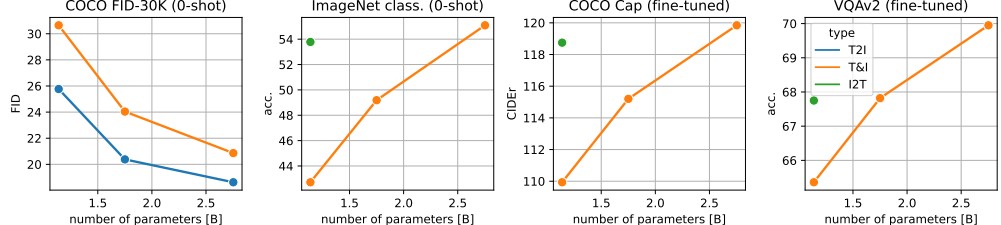

Figure 4: Zero-shot image sample quality, zero-shot ImageNet classification accuracy (by scoring class labels, without prompts), and captioning and VQA performance as a function of model size for JetFormer variants. Increasing the model size improves all the metrics. T&I models generally perform worse than unidirectional models which are trained either for T2I or I2T generation.

Table 4: Comparison with baselines pretrained on raw image-text data and fine-tuned for image captioning (COCO captions; CIDEr) and VQA (VQAv2; test-dev acc.). Models marked with T&I were jointly pretrained for captioning and T2I generation. *The numbers are from (Tschannen et al., 2023) for models pretrained on 900M image-text pairs, which most closely matches our setup.

|  | extra step | COCO cap. | VQAv2 |
|---|---|---|---|
| CapPa L/14 (Tschannen et al., 2023)* | – | 118.7 | 68.6 |
| CLIP L/14 (Radford et al., 2021)* | – | 118.2 | 67.9 |
| ARGVLT (T&I) (Kim et al., 2023) | VQ-VAE | 94.7 | – |
| MAGVLT Large (T&I) (Kim et al., 2023) | VQ-VAE | 110.7 | 65.7 |
| JetFormer-B (I2T) | – | 118.7 | 67.2 |
| JetFormer-L (T&I) | – | 119.8 | 70.0 |

## 5 CONCLUSION

In this paper we introduce JetFormer, a novel class of generative models that combines normalizing flows and autoregressive models with soft tokens. To the best of our knowledge, it is the first image model, which is capable to synthesize high-resolutions images and provide an explicit (and competitive) NLL bounds for the raw images. JetFormer is a fully end-to-end trainable model (with no components pretrained ahead of time), which means that it can be fully tailored towards the task at hand, without being limited by the external and frozen components. Being able to compute NLL is also an important feature of our model. NLL is a tangible score closely related to compression capabilities, and can be used to compare various generative models across different modeling classes or for hill-climbing. Also, by measuring NLL score one can ensure the absence of the mode collapse, because mode collapse will lead to deterioration NLL on the hold-out data.

We note that JetFormer in its current form also has some limitations. The visual quality of its samples lags behind state-of-the-art diffusion models that leverage pretrained latent representations. Additionally, the full end-to-end nature of JetFormer also comes with increased computational requirements. However, given JetFormer's simple design, we believe that it can be scaled up well so that the benefits of end-to-end training can come to full fruition.

**Reproducibility Statement**  We provide detailed information about the training recipe, the architecture, hyper-parameters and the training data in Section 4 and Appendix A.

**Ethics Statement**  This paper describes a system for understanding and generation of image-text data, with focus on characterizing and exploring its performance on academic data sets. It fits into the broader class of large multimodal models trained on data from the web, and the same ethical implications as for those prior works apply here. In particular, when releasing or deploying such models publicly, extensive measures to de-biasing the data should be taken, and models should be safety-tuned and red-teamed prior to release. Content-filters can be added to the inference pipeline to further improve safety. We refer to the corresponding papers for a more in-depth discussion, for example (Radford et al., 2021; Chen et al., 2023a; Po et al., 2024).

ACKNOWLEDGEMENTS

We thank Lucas Beyer for a detailed review and feedback on the final manuscript. We further thank Joan Puigcerver for timely infrastructure contributions.

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

APPENDIX – SUPPLEMENTARY MATERIAL

## A ARCHITECTURES AND HYPERPARAMETERS

Table 5 shows the shapes of the autoregressive transformer and normalizing flow components for different JetFormer variants. One can see that the dense layer predicting the GMM parameters are relatively large. To save memory, we store this layer in the `bfloat16` format (instead of `float32` as the other parameters) which does not impact the quality. Also note that at inference time applying this layer can be greatly sped up by first sampling the mixture and then only inferring the mean and covariance of that mixture.

We rely on the design from (Kolesnikov et al., 2024) for the normalizing flow, which uses a stack affine coupling blocks (Dinh et al., 2016, Eqn. 7, 8). For each block $i$ the relation between input sequence $y_i$ and output $y_{i+1}$ is

$$\bar{y}_{i+1} = \bar{y}_i \qquad\qquad \Leftrightarrow \qquad \bar{y}_i = \bar{y}_{i+1},$$
$$\tilde{y}_{i+1} = (\tilde{y}_i + b_i(\bar{y}_i)) \odot \sigma(a_i(\bar{y}_i)) \qquad \Leftrightarrow \qquad \tilde{y}_i = \tilde{y}_{i+1}/\sigma(a_i(\bar{y}_{i+1})) - b_i(\bar{y}_{i+1}).$$

Here, $\bar{y}_i$ and $\tilde{y}_i$ are obtained by splitting $y_i$ into equally-shaped halves along the channel dimension according to a randomly initialized partition (and $y_{i+1}$ results from merging $\bar{y}_{i+1}$ and $\tilde{y}_{i+1}$ according to this partition). The scale $a_i$ and shift $b_i$ are inferred by two separate heads from a shared ViT $g_i$.

The normalizing flow component has about 450M to 650M parameters, depending of the variant. This is more than typical CNN-based (VQ-)VAEs for image generation in the literature (Esser et al., 2020; Rombach et al., 2022), but comparable to common ViT-based VQ-VAEs (Yu et al., 2022a;c).

Table 6 shows the hyper-parameters used to fine-tune JetFormer. Overall we observed dropout to have a significant impact on the image sample quality. When reporting fine-tuning-based metrics we report the median of 10 runs, except in VQAv2 where we report test-dev accuracy from the evaluation server.

Table 5: Architecture details for different JetFormer variants. See Section A for a discussion.

|  | JetFormer-B | JetFormer-M | JetFormer-L |
|---|---|---|---|
| *Autoregressive transformer* | | | |
| Depth | 24 | 24 | 48 |
| Width | 1024 | 1536 | 1536 |
| MLP hidden dim. | 4096 | 6144 | 6144 |
| Num. heads | 16 | 16 | 16 |
| Num. KV heads | 1 | 1 | 1 |
| Head dim. | 64 | 96 | 96 |
| Vocab. size | 32k | 32k | 32k |
| Num. mixtures | 1024 | 1024 | 1024 |
| Num. GMM param. | 269M | 404M | 404M |
| Num. vocab. param. | 33M | 49M | 49M |
| Num. backbone param. | 389M | 847M | 1.65B |
| *Normalizing flow* | | | |
| Num. coupling blocks | 32 | 32 | 32 |
| Block width | 512 | 512 | 512 |
| Block depth | 4 | 4 | 6 |
| Block MLP hidden dim. | 2048 | 2048 | 2048 |
| Block num. heads. | 8 | 8 | 8 |
| Num. param. | 447M | 447M | 650M |
| Total num. param. | 1.38B | 1.75B | 2.75B |

## B FACTORING OUT REDUNDANT DIMENSIONS: PCA-INSPIRED VARIANT

Recall that for this variant, before feeding $x$ to the flow model, we reshape it into a sequence of flattened patches and apply a learnable, invertible linear map $W$ along the channel dimension.

Table 6: Hyper-parameter details for fine-tuning tasks.

| Task | Model Size | Epochs | Batch size | Learning rate | Weight decay | Dropout | Sampler |
|---|---|---|---|---|---|---|---|
| COCO (FID-30k) | B/M/L | 10 | 256 | 1e-4 | 1e-5 | 0.1 | CFG |
| VQAv2 | B/M/L | 10 | 128 | 3e-5 | 3e-6 | 0.0 | Greedy |
| COCO caption | B/M | 10 | 128 | 3e-5 | 3e-6 | 0.0 | Greedy |
| COCO caption | L | 5 | 256 | 3e-5 | 3e-6 | 0.0 | Greedy |

For this variant, the loss can be written as:

$$L(x) = \log p(f(\hat{x})) + \log p_{\mathcal{N}}(\tilde{x}) + \log \left| \det \left( \frac{\partial f(\hat{x})}{\partial \hat{x}^T} \right) \right| + (HW/p^2) \log |\det W|,$$

where $p_{\mathcal{N}}$ is the Gaussian distribution. $xW^T$ is decomposed into two slices as $[\hat{x}, \tilde{x}] = xW^T$ of shapes $HW/p^2 \times d$ and $HW/p^2 \times (3p^2 - d)$, respectively, where the original image has shape $H \times W \times 3$ and is split it into $p \times p$ patches which are then flattened.

Intuitively, factoring out dimensions via $W$ rather than at the flow output limits the complexity and hence potentially leads to worse results (see Table 2).

When compting $W$ via PCA to obtain the bottom two rows in Table 2, we sample $4\,000$ images from the ImageNet training set, split them into patches, and compute PCA using the built-in implementation from `scikit-learn` without whitening.

## C  ADDITIONAL ABLATIONS

**VAE baseline**   We train a VAE following the design of VQGAN (Rombach et al., 2022) producing a sequence of 256 128-dimensional tokens like the normalizing flow in JetFormer. We remove the GAN and perceptual losses to ensure a fair comparison with JetFormer (which solely maximizes the data likelihood, without relying on GAN or perceptual losses). We then train the JetFormer-B decoder-only model on this VAE representation, following the JetFormer-B T&I training recipe. The results in the Table 7 below show that JetFormer outperforms the VAE baseline by a solid margin (in particular in T2I generation).

Table 7: Comparison of JetFormer-B trained for T&I with a 2-stage VAE-based variant.

| | COCO-FID30k | INet 0-shot | COCO Cap | VQAv2 |
|---|---|---|---|---|
| VAE + Transformer (2 stages) | 95.4 | 37.6 | 103.9 | 64.0 |
| VAE + Trans. (2 stages, w/ noise curr.) | 87.6 | 40.0 | 106.7 | 64.9 |
| JetFormer (end-to-end) | 30.6 | 42.7 | 113.9 | 65.4 |

**Effect of noise curriculum on NLL**   Intuitively, the noise curriculum biases the training process towards those high-likelihood solutions with high perceptual quality, at the expense of a slight increase in NLL (see Sec. 3.2.1 for more discussion). Table 8 shows that longer training with noise curriculum eliminates the gap in NLL compared to training without curriculum.

Table 8: Sample quality and NLL as a function of noise curriculum and training duration.

| noise curr. | #epochs | FID-50k | NLL |
|---|---|---|---|
| No | 100 | 44.71 | 3.05 |
| Yes | 100 | 7.84 | 3.14 |
| Yes | 500 | 7.25 | 3.06 |
| Yes | 1000 | 7.10 | 3.04 |

**Effect of final noise scale when fine-tuning for T2I generation**    We further visualize the interplay between the final noise scale and NLL when fine-tuning T2I models on COCO in Fig. 5.

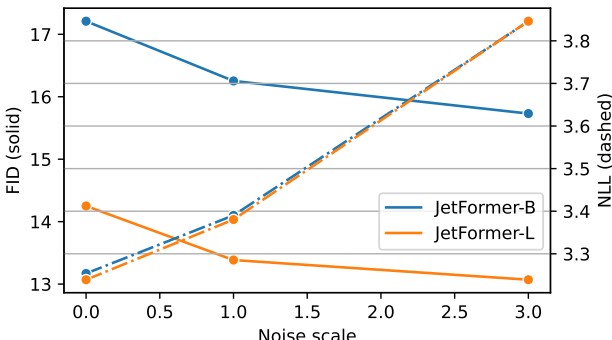

Figure 5: FID and NLL results obtained when fine-tuning JetFormer for T2I generation on COCO with different values for the final noise scale. JetFormer is pretrained with a curriculum decaying the noise scale from 64 to 3 and the noise scale is then decayed during fine-tuning from 3 to one of $\{0, 1, 3\}$. We observe that decaying the noise scale to 0 leads to the best NLL but a worst FID. The two model sizes have similar NLL at noise scale 3 but different FIDs, possibly because NLL is bounded by the noise scale but larger model scale helps to generalize better.

## D  IMAGE AND LATENT MANIPULATION

In Figure 6 we visualize the effect of different image or latent manipulations. Note that *these images are not indicative of the final image generation quality* and their purpose is to illustrate certain qualitative properties. We use four images: a full $256{\times}256$ view of a plane from ImageNet (Russakovsky et al., 2015), a $64{\times}64$ crop of an image of a bird from ImageNet, a $64{\times}64$ crop of a scene from MS-COCO (Lin et al., 2014) to analyze out-of-distribution generalization (vs. ImageNet), and a $128{\times}128$ crop of an image with text from TextVQA (Singh et al., 2019), to visualize the effect of a significantly different domain (vs. natural images). On those images we show:

(a) Ground truth.

(b) Additive RGB noise $N(0, \mathrm{I})$ with scale $\sigma_0 = 64$. This corresponds to the images at the beginning of the noise curriculum. Although this significantly affects the perception of the zoomed in view of the bird or details of the clouds, it has smaller impact on the more zoomed out features such as the plane at $256{\times}256$ or the large text letters.

(c) In Section 3.2 we discussed factoring out redundant dimensions. Here we use a model trained on ImageNet where the dimensions are factored out after the normalizing flow to map the images to the latent space. After we keep the 128 autoregressive dimensions fixed but resample the 640 dimensions modelled by the Gaussian prior. Despite these corresponding to 640 of 768 latent dimensions of each patch, they have little impact on the overall image structure. It shows the model successfully maps important information to the 128 dimensions modeled with the autoregressive prior which are unmodified in this manipulation.

(d) Zeroing out the gaussian latents after a learnable projection. Overall the model learned to map the relevant dimensions into the 128 per patch modelled further by normalizing flow.

(e) PCA reconstruction of patches when using 128 dimensions. PCA is from ImageNet images.

(f) VAE encoding and decoding. We use a VAE from (Tschannen et al., 2024) which was pretrained on ImageNet with perceptual losses, overall it produces sharp images but looses significant detail and is not able to preserve details in out-of-distribution images, *e.g.* with text.

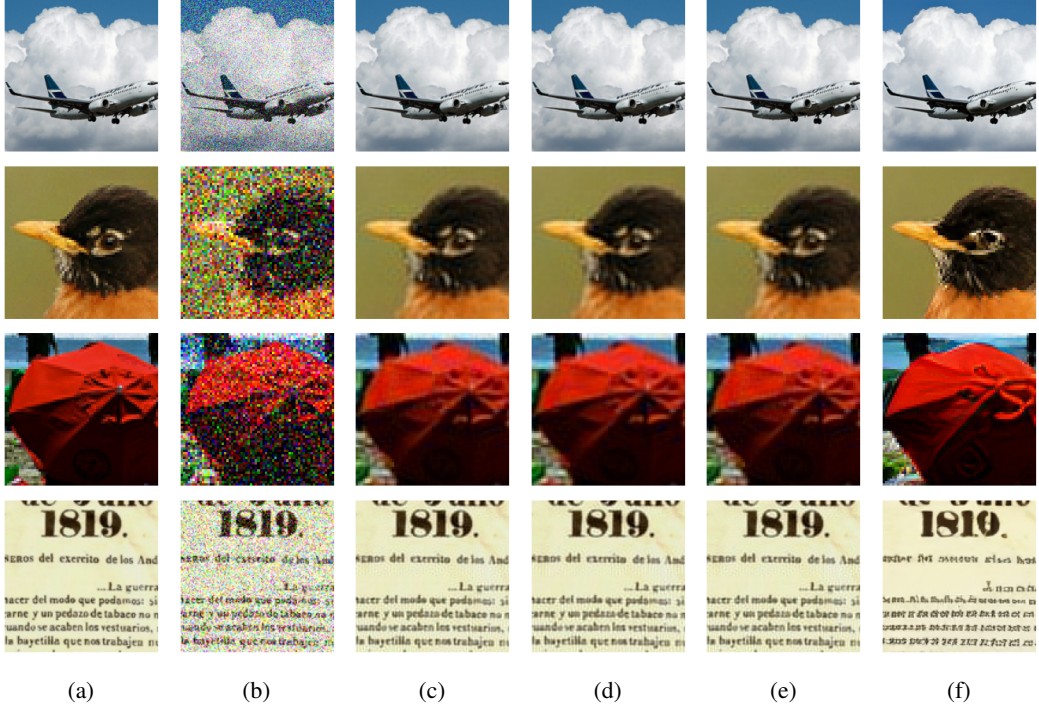

Figure 6: Visualization of image and latent manipulations. Note *these images are not indicative of the final image generation quality*. Best viewed in digital format.

# E   ZERO-SHOT MS-COCO SAMPLES

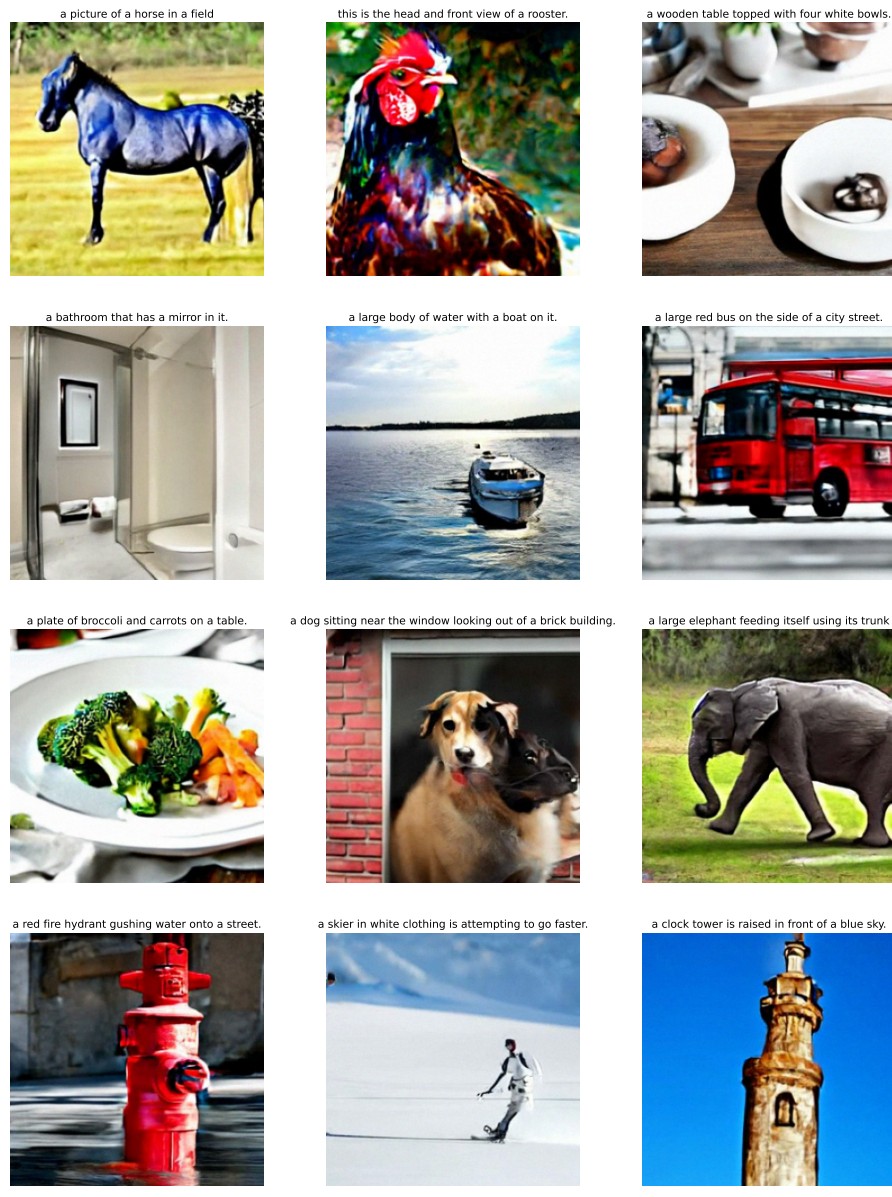

Figure 7: Selected zero-shot samples from JetFormer-L (T2I) for captions from MS-COCO.

# F    IMAGENET SAMPLES

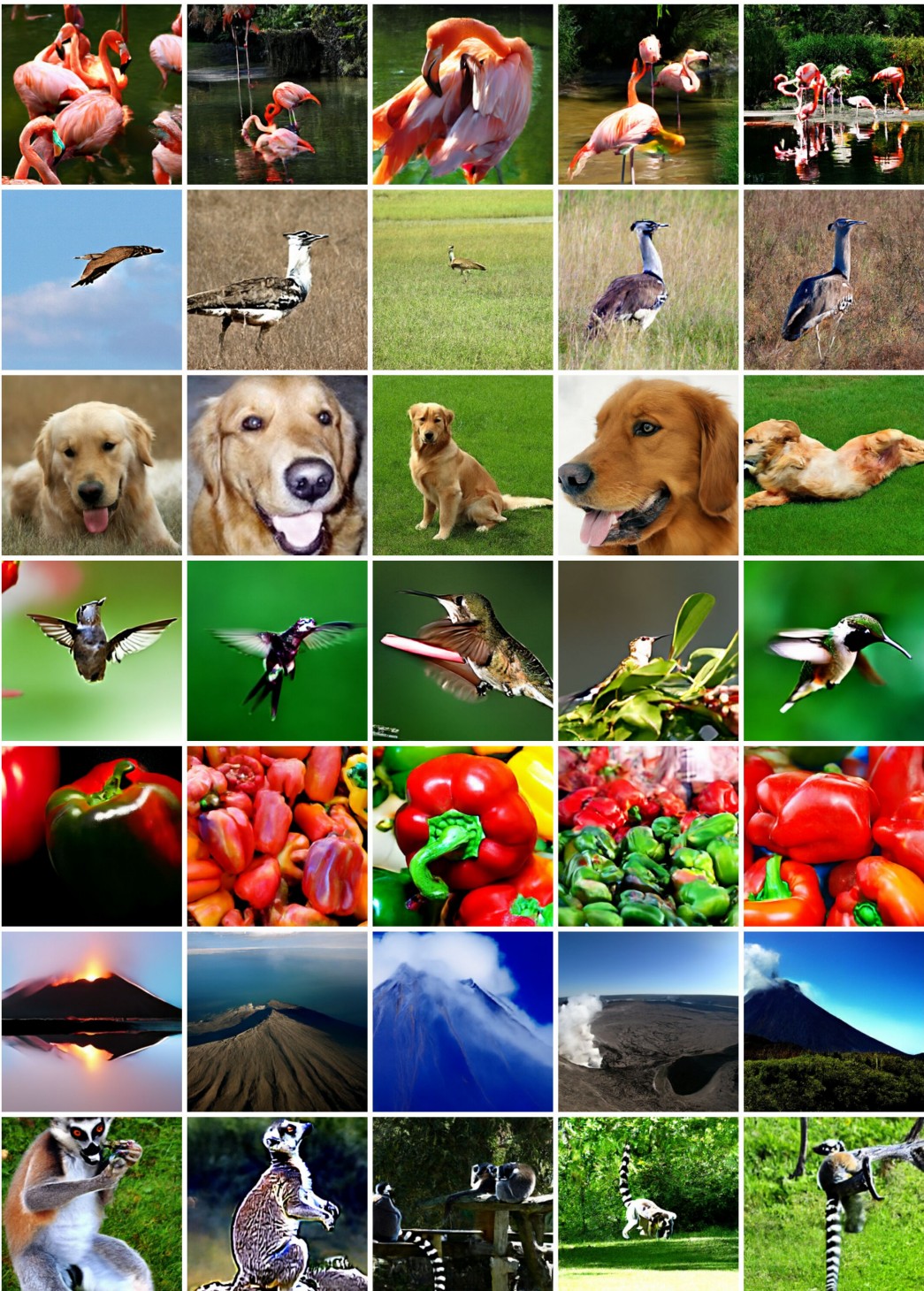

Figure 8: Random, non-cherry picked samples from JetFormer-L for selected ImageNet classes: flamingo, bustard, golden retriever, hummingbird, bell pepper, volcano and ring-tailed lemur.

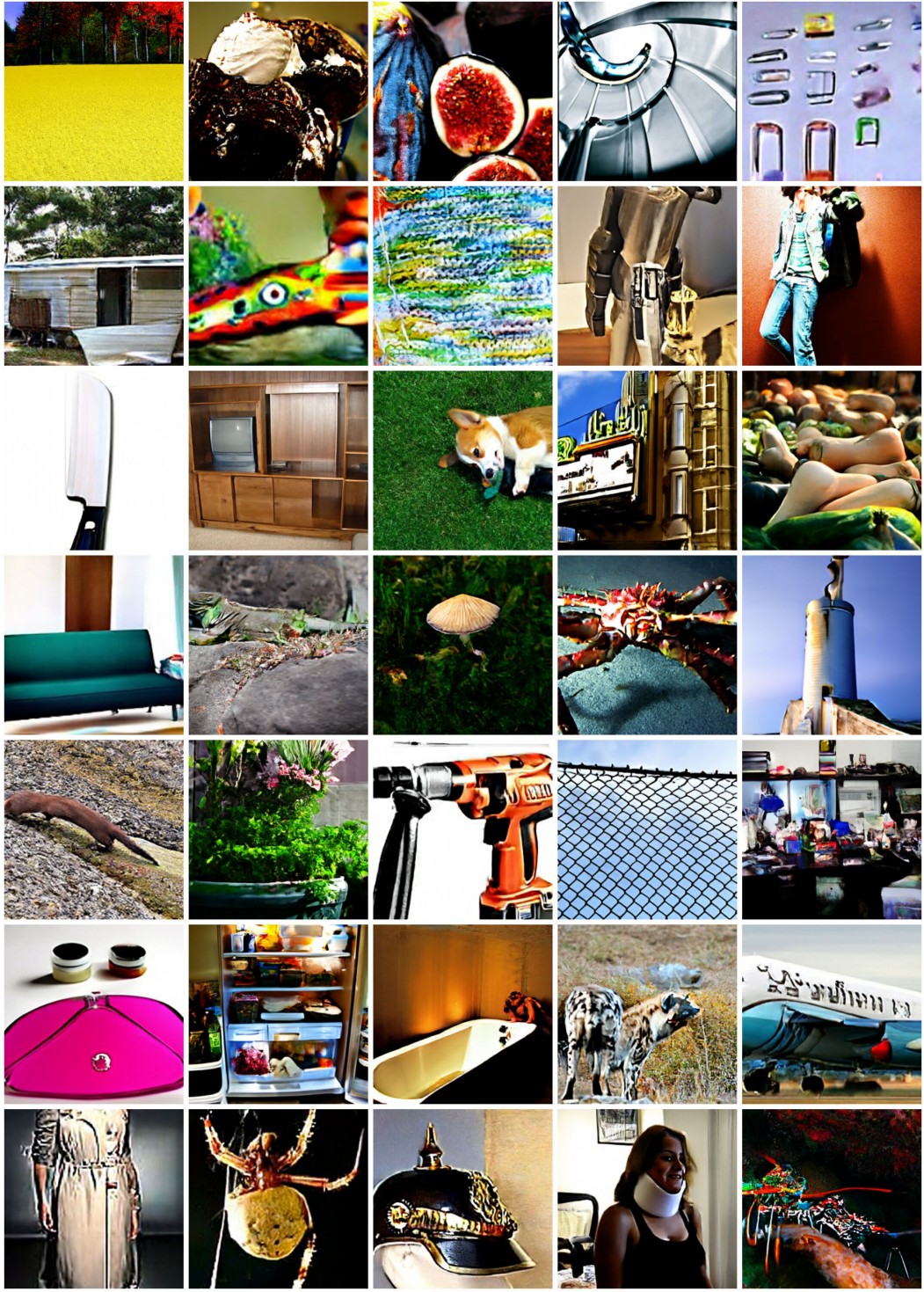

Figure 9: Random, non-cherry picked samples from JetFormer-L for random ImageNet classes.

