# OpenReview forum: "JetFormer: An autoregressive generative model of raw images and text"
_ICLR.cc/2025/Conference — ICLR 2025 Poster_

### Official Review · Reviewer_NBCt · 2024-10-30

**Soundness:** 2
**Presentation:** 3
**Contribution:** 2
**Rating:** 5
**Confidence:** 4

**Summary:**

This paper propose a generative model for generating text and images, it propose to use a normalizing flow model to simultaneously encode and decode images. The flow model generates soft token image representation that is jointly trained with an autoregressive decoder only transformer architecture.

The main contribution of the paper is to first explore the usage of normalizing flow models as image tokenizers that can achieve an end-to-end training without the need to use a pretrained autoencoder for images.

**Strengths:**

- The paper is well written and easy to follow.

- The exploration of a normalizing flow model in vision language pretraining setting is an original and unexplored idea.

- The proposed method eliminate the needs to rely on pretrained components, by training the whole system end-to-end, the use of soft-tokens with normalizing flow is needed to avoid the mode collapse typical of VAEs.

**Weaknesses:**

- The central claims of the paper are not supported by experimental results: The model performance on both text-to-image taskt is worse with respect to older and smaller models based on pretrained VQ-VAE (e.g MAGVLT [1]), while the improvement on understanding tasks is slight, training for image understanding greatly reduces the performance on image generation.

- The paper does not compare with recent baselines such as VAR models [2] for simple settings like ImageNet256 where VAR achieve 1.92 FID and Jetformer has 6.64 using the same training data and similar model size.

- Optimizing image tokens during training can add instability to the training dynamics, this is especially true in complex large scale multimodal settings.

- The main comparison to validate the usage of this modeling approach with respect to use a VQ-VAE in a controlled setting would be to compare it against a baseline, trained in the exact same setting  by substituting the normalizing flow with a VQ-VAE.

- How the approach scales is not clearly analized in the paper by just showing to models scales with relative low difference in parameters count

[1]: MAGVLT: Masked Generative Vision-and-Language Transformer

[2]: Visual Autoregressive Modeling: Scalable Image Generation via Next-Scale Prediction, Tian et al.

**Questions:**

- What do you think it is the main practical advantage of removing the pretrained image tokenizer for autoregressive image modeling?

- Recent papers as Transfusion [3] shows strong results by training with mixed modal corpus of data, a discussion on how this approach compare to your method would be beneficial.

- Recent developments in image generation and deep learning guided compression shows that using a pretrained autoencoder to compress and reduce image dimensionality is really effective, moreover decoupling image tokenization from the generative training strategy can have some benefits in terms of training dynamics. What is your perspective on these considerations?


[3] Transfusion: Predict the Next Token and Diffuse Images with One Multi-Modal Model. Zhou et al

---

> ### Author Response · Authors · 2024-11-23
>
> We thank the reviewer for their thoughtful comments, which we address in the following.
>
> **Comparison with MAGVLT (T2I) and and VAR (ImageNet246):** JetFormer is a pixel-likelihood based model trained end-to-end from raw pixels, whereas MAGVLT, VAR (and related models) are two-stage models, whose first stage consists of learning a semantic, abstract representation, which requires a complex mix of reconstruction, perceptual and GAN losses (and explicit multi-scale decomposition in VAR). Likelihood maximiziation only happens in the second stage, in an abstract, semantic space optimized for human perceptual quality. MAGVLT and VAR hence belong to a different model class which explicitly optimizes for perceptual quality (concretely via LPIPS derived from user rating in VQ-VAE training). Compared to the best prior pixel likelihood-based image generation methods such as Image Transformer (Parmar et al. 2018) and iGPT (Chen et al. 2020), JetFormer’s image generation capabilities are a large step forward in terms of resolution and visual quality, and this at computational requirements comparable with VQGAN.
>
> **Training instability:** We did not observe training instabilities or similar issues. In fact, end-to-end training of the normalizing flow and the transformer decoder is fully differentiable and does not change the training objective (data likelihood) but merely adds a log-determinant term (see Sec. 3.1). In particular the log-determinant term is analytical and does not have a weight associated, and hence does not lead to additional tuning requirements.
>
> **Comparison with VAE-based approach:** Thank you for this suggestion. We train a VAE (not a VQ-VAE, to stay as close as possible to JetFormer’s soft token based approach) following the design of VQGAN/MaskGIT. The VAE produces a sequence of 256 128-dimensional tokens like the normalizing flow in JetFormer. We remove the GAN and perceptual losses to ensure a fair comparison with JetFormer (which solely maximizes the data likelihood, without relying on GAN or perceptual losses). We then train the JetFormer-B decoder-only model on this VAE representation, following the JetFormer-B T&I training recipe. The results in the table below show that JetFormer outperforms the VAE baseline by a solid margin (in particular in T2I generation).
>
> |                               | COCO-FID30k | INet 0-shot | COCO Cap | VQAv2 |
> |-------------------------------|:-------------:|:-------------:|:----------:|:-------:|
> | VAE + Transformer (2 stages)  | 95.4        | 37.6        | 103.9    | 64.0    |
> | VAE + Transformer (2 stages, w/ noise curriculum)      | 87.6        | 40.0          | 106.7    | 64.9  |
> | JetFormer (end-to-end)        | 30.6        | 42.7        | 113.9    | 65.4  |
>
>
> **Scaling behavior:** Besides extensive ablations on design choices, we spent significant compute to cover transformer decoder sizes ranging GPT-3 M to XL size or a factor of 4x in model size in our multimodal experiments, which we consider enough to validate a first generation, novel model. Indeed, we see the zero-shot metrics vary 20+% in this range.
>
> **Comparison with Transfusion:** We briefly discuss transfusion in the related work section of the original submission. While we believe transfusion is a very interesting approach, it also uses a pretrained VAE and does not directly operate on pixels. Furthermore, its objective is not pure data likelihood maximization, but a combination of next-token prediction (text) and diffusion denoising objective (images). JetFormer differs significantly from Transfusion in these two aspects.

---

> ### Author Response · Authors · 2024-11-23
>
> **Practical advantages of removing the tokenizer:** The limitations of pre-trained, modality-specific encoders in the context of multimodal LLMs are well documented and quantified in the literature, see for example [1, 2, 3]. Specifically, [1, Fig. 2] illustrates a clear tradeoff for between T2I and I2T model performance when varying the either number of T2I or I2T training examples, while keeping the number of I2T and T2I training examples fixed, respectively. The solution proposed in [1] to avoid this tradeoff is to use decoupled representations for understanding and generation.
>
> In a broader context, JetFormer is the first model which enables multimodal generative pretraining on raw data via next-token prediction by solely optimizing the data likelihood, akin to pretraining ubiquitous in LLMs. Besides avoiding specialization of the image representation (and its limitations), this also enables architecture-agnostic comparison of validation likelihoods (and hence hill-climbing) and scaling laws (as those do not depend on a specific visual tokenizer used).
>
> Overall, while the reviewer raises valid points in favor of the practical advantages of VQ-VAEs when used carefully, we would like to point out that JetFormer is a first of its kind approach and expecting it to beat VQ-based approaches based on 100s or 1000s of incremental improvements from prior publications is an extremely high bar. We believe fully end-to-end paradigms like JetFormer could come to full fruition in the future if the current scaling trends in compute and unification across modalities continues.
>
> [1] Pan, Kaihang, et al. "Auto-Encoding Morph-Tokens for Multimodal LLM." ICML 2024 \
> [2] Xu, Yifan, et al. "Libra: Building Decoupled Vision System on Large Language Models." ICML 2024 \
> [3] Dong, Runpei, et al. "DreamLLM: Synergistic Multimodal Comprehension and Creation." ICLR 2024

---

> > ### Comment · Reviewer_NBCt · 2024-11-25
> > **Response to Rebuttal**
> >
> > I thank the authors for their response and for addressing my feedback. I have mixed-feelings about this work, on one hand it pioneers a new approach, on the other hand, experimental results doesn't seem to be convincing enough to support the introduced paradigm. For these reasons I will keep my score.

---

> > > ### Author Response · Authors · 2024-11-27
> > >
> > > We thank the reviewer for their response and would like to encourage reconsideration of the score. In particular, we highlight that according to point 4 of the reviewer guidelines [1], a submission can be considered significant to the community without achieving state-of-the-art results.
> > >
> > > This work trains a generative autoregressive model on raw images and text end-to-end, following the LLM pretraining paradigm via next token prediction, without relying on pretrained image representations like VQ-VAE. To our knowledge, no other methods enable this type of pretraining.
> > >
> > > JetFormer achieves solid ImageNet256 results, competitive with VQGAN (Esser et al., 2020), introduced three years after VQ-based generative models (van den Oord et al., 2017), and surpasses early VQ-based models like DALL-E (Ramesh et al., 2021) in T2I generation, even with simultaneous I2T training. We also provided the requested ablations, showing end-to-end training clearly outperforms the 2-stage baseline, without perceptual losses or advanced tricks.
> > >
> > > [1] https://iclr.cc/Conferences/2025/ReviewerGuide

---

> > > > ### Comment · Reviewer_NBCt · 2024-11-28
> > > > **Response by Reviewer**
> > > >
> > > > I thank the authors for engaging in the discussion, the reason of my score (slighlty lower than the acceptance bar) is that I am not convinced by the experimental results in support of the claims of the paper (i.e. I cannot see a clear advantage in jointly learning the image tokenizer and the autoregressive model. I agree with the authors about achieving state-of-the-art results and the fact that this paper is not SOTA does not affect the score in any way.
> > > >
> > > > According to point 3 of the reviewer guidelines mentioned by the authors, the question "Does the paper support the claims?" does not have a clear answer from my perspective: I think the paper showcase a new methodology which is not clearly supported by the experimental evidence. Moreover, I see the fact that decoupled and pretrained image tokenizers represent images in a task-agnostic way as an advantage to be more generalizable across a variety of tasks, in contrast with the authors view.

---

### Official Review · Reviewer_Lv3i · 2024-11-02

**Soundness:** 4
**Presentation:** 3
**Contribution:** 3
**Rating:** 8
**Confidence:** 4

**Summary:**

The paper proposes an end-to-end autoregressive model that can jointly understand and generate both image and text. The model consists of two components. A normalizing flow is applied to convert images into soft tokens for autoregressive modeling. The main backbone is a decoder-only transformer that predicts the next soft token. The paper explores two effective approaches to help model convergence. One is to introduce an evolving noise level during the training schedule. Another one is to dropout the redundant dimensions within the soft token when modeling the distribution auto-regressively. Experiments are conducted to evaluate the proposed method both quantitatively and qualitatively.

**Strengths:**

1. The paper is well-writen and is easy to follow.
2. The proposed method is innovative, showing a promising approach towards end-to-end image-text joint probabilistic modeling. It shows that it is possible to directly optimize NLL for potentially any modality.
3. The paper identifies two crucial training techniques to help the model converge better, which are crucial for the method's effectiveness.
4. Experiments are conducted sufficiently. It also shows the proposed method can potentially benefit from scaling up the model size.

**Weaknesses:**

1. In figure 3 of the paper, it seems that the noise curriculum makes the final NLL get slightly higher than without using the noise curriculum, but experimentally, the model clearly benefits from the noise curriculum training technique. Does this indicate that NLL may not be the best objective for image-text generation task? A more theoretical explanation is helpful for the readers to understand this phenomenon.
2. The detailed architecture of the normalizing flow model is not clearly explained. Better make it clear by showing diagrams of the model blocks and calculations.

**Questions:**

1. About information leakage. Since the image tokenizer is also trainable, it seems possible for the image tokenizer to attend to the next input token that the auto-regressive transformer should predict, i.e. the information of the next token is leaked into the input tokens, causing a trivial solution. Just wonder why this won't happen in Jetformer?

---

> ### Author Response · Authors · 2024-11-23
>
> We thank the reviewer for their thoughtful comments, which we address in the following.
>
> 1. **NLL vs. visual quality:** This is an interesting observation. Our interpretation is that the noise curriculum biases the training process towards those high-likelihood solutions with high perceptual quality, at the expense of a slight increase in NLL. As discussed in Sec. 3.2.1 (and in (Kingma & Gao, 2023; Dieleman, 2024) for related models) the noise curriculum encourages learning image generation in a coarse-to-fine manner. This is somewhat similar in spirit to post training in LLMs: The raw LLM trained via next-token prediction and NLL minimization does not necessarily produce outputs useful for humans, but slightly altering the model weights via post-training makes the results very useful. \
> We further note that longer training with noise curriculum eliminates the gap in NLL compared to training without curriculum (results for JetFormer-B on ImageNet256):
>
> | noise curriculum | #epochs | FID-50k | NLL  |
> |------------------|---------:|---------:|------:|
> | No               | 100     | 44.71   | 3.05 |
> | Yes              | 100     | 7.84    | 3.14 |
> | Yes              | 500     | 7.25    | 3.06 |
> | Yes              | 1000    | 7.10     | 3.04 |
>
> 2. **Normalizing flow architecture:** Our design follows the affine coupling layers from (Dinh et al. 2017). The only difference is that we use stacks of ViT blocks, specified in Table 5 (bottom section) of the original submission, instead of CNNs to compute the couplings. We added the forward and backward computation (Eqns. 7 and 8 in (Dinh et al. 2017)) in Appendix A of the paper. Together, this provides a complete characterization of the normalizing flow architecture.
>
> **Information leakage in end-to-end training:** Information leakage is not possible by model design. The flow model is information preserving (invertible, bijective) and thus cannot collapse or “cheat” in other ways. For example, aggregating all the information in a subset of the tokens (for example the first token) is not possible as this would imply the flow is not bijective. Yes, it can in principle organize information in tokens in a peculiar way, e.g. anti-causal, but it won’t result in any net gains: the overall JetFormer model will obtain a lower NLL (objective value) if the information is organized in a way such that the transformer decoder can model the latents well.

---

> > ### Comment · Reviewer_Lv3i · 2024-11-26
> >
> > The author's response resolved my issue, and I have decided to maintain my rating.

---

### Official Review · Reviewer_jKRf · 2024-11-02

**Soundness:** 3
**Presentation:** 3
**Contribution:** 3
**Rating:** 6
**Confidence:** 3

**Summary:**

The paper titled "JetFormer: An autoregressive generative model of raw images and text" presents a novel system designed for the understanding and generation of image-text data, with a particular focus on its performance on academic datasets. The model, JetFormer, is part of a broader category of large multimodal models trained on web data, and it shares similar ethical considerations, such as the need for de-biasing, safety-tuning, and the use of content filters to ensure safe deployment.

JetFormer is characterized by its end-to-end training approach, which, while computationally demanding, offers the potential for scalability and improved performance. However, the model currently faces limitations in visual quality compared to state-of-the-art diffusion models that utilize pretrained latent representations. Additionally, the computational requirements are significant, especially when dealing with high-resolution images, which necessitates efficient handling of image data, such as modeling at the patch level.

The paper also discusses a PCA-inspired variant to reduce redundant dimensions, which involves reshaping input data into sequences of flattened patches and applying a learnable, invertible linear map. This approach aims to manage the complexity of the data and improve the model's efficiency.

Overall, the contributions of the paper include the development of JetFormer, an exploration of its capabilities and limitations, and the proposal of methods to enhance its performance and safety in practical applications.

**Strengths:**

Originality: The paper introduces innovative techniques to enhance the quality of image generation models, particularly through the use of a novel noise curriculum and the factoring out of latent dimensions. The approach of factoring out redundant dimensions using a PCA-inspired variant before applying a flow model is a creative solution to improve model efficiency and performance. This originality is further demonstrated by the introduction of classifier-free guidance during sampling, which adds a unique aspect to the model's design.

Quality: The quality of the research is evident in the thorough experimentation and evaluation conducted. The paper employs robust evaluation metrics, such as the ADM FID evaluation suite and precision/recall metrics, to assess image sample quality on ImageNet1k. Additionally, the use of MS-COCO FID-30k for text-to-image tasks and the reporting of CIDEr scores for image captioning and VQAv2 accuracy for visual question answering further demonstrate the comprehensive nature of the quality assessment. The results indicate that the proposed enhancements significantly improve the quality of generated images.

Clarity: The paper is well-structured and clearly articulates the methodologies and findings. The explanation of the process of factoring out redundant dimensions and the subsequent impact on model performance is detailed and easy to follow. The use of visual aids or diagrams, if included, would further enhance the clarity of the presentation, but the textual explanation alone provides a solid understanding of the research contributions.

Significance: The significance of the paper lies in its potential impact on the field of image generation and understanding. By addressing the computational challenges associated with modeling images at the patch level and proposing solutions that reduce computational burden while maintaining or improving image quality, the research offers valuable insights and tools for future developments in the field. The ability to improve model quality for natural images and the implications for tasks such as zero-shot classification, image captioning, and visual question answering highlight the broad applicability and importance of the work.

**Weaknesses:**

Model Performance and Design Choices: The paper highlights several design choices that impact model performance, such as the use of dropout and PCA transforms. It notes that modeling images after PCA leads to worse results and that omitting noise curriculum results in significantly poorer outcomes. These observations suggest that the model's performance is sensitive to specific preprocessing and design choices, which may limit its robustness and generalizability.

Factoring Dimensions: The paper discusses the impact of factoring out dimensions at different stages of the model. It finds that not factoring out dimensions after a flow model leads to quality loss, and while factoring out dimensions with a learnable linear mapping before the flow improves results, it still underperforms compared to post-flow factoring. This indicates a potential area for improvement in the model's architecture to optimize performance.

Lack of Comprehensive Evaluation: While the paper reports metrics based on the median of 10 runs and test-dev accuracy for VQAv2, it does not provide a thorough evaluation across diverse datasets or scenarios. A more extensive evaluation would help in understanding the model's strengths and limitations better.

Addressing these weaknesses would enhance the paper's contribution to the field and provide a clearer understanding of the model's capabilities and limitations.

**Questions:**

Please refer to the weaknesses.

---

> ### Author Response · Authors · 2024-11-23
>
> **Factoring out dimensions and design choices:** We would like to propose an alternative angle to interpret our investigation on factored-out dimensions, PCA and invertible projection. Natural images have a large amount of redundancy (or, in other words, are strongly compressible). This makes it wasteful and challenging to model raw image data. The classical approach is to use domain-specific tokenizers, which were pretrained to reduce the redundancy and help a generative model to focus on modeling non-redundant information. \
> Being a powerful end-to-end model, JetFormer allows to handle such redundancy in a principled manner. Our main proposed strategy is to factor out dimensions after the normalizing flow. The flow is incentivized to reduce (zero-out) the factored-out dimensions. Due to the importance of this topic, we additionally demonstrate other ways to handle redundancy: via PCA or invertible dense projection, which also show promising results, but are not as effective as the former approach. \
> Overall, we argue that JetFormer enables redundancy modeling in a principled way, and also allows for interesting alternatives that also perform quite well.
>
> **Lack of Comprehensive Evaluation:** Our main contribution is to propose the first model which enables multimodal generative pretraining on raw data via next-token prediction by solely optimizing the data likelihood, akin to pretraining ubiquitous in LLMs. Compared to the best prior pixel likelihood-based image generation methods such as Image Transformer (Parmar et al. 2018) and iGPT (Chen et al. 2020), JetFormer’s image generation capabilities are a large step forward in terms of resolution and visual quality, and this at computational requirements comparable with VQGAN. While we do not perform extensive post-training and associated evaluations involving e.g. multi-turn interactions and text-guided editing, our evaluations show that JetFormer can learn basic multimodal capabilities by next-token prediction on raw data. We hence leave further scaling and advanced post training with associated evaluations for future work.

---

> > ### Comment · Reviewer_jKRf · 2024-11-25
> >
> > The author's response resolved my issue, and I have decided to maintain my rating.

---

### Official Review · Reviewer_9Pqq · 2024-11-05

**Soundness:** 3
**Presentation:** 3
**Contribution:** 2
**Rating:** 6
**Confidence:** 3

**Summary:**

This paper proposes JetFormer, an end-to-end autoregressive transformer designed for multimodal generation of text and images. JetFormer utilizes a normalizing flow to encode images into soft tokens, eliminating the need for pretrained encoders and enabling direct learning from raw data. The model’s unified structure for text and image processing enhances its multimodal adaptability, with additional techniques like a noise curriculum to improve image quality. JetFormer achieves competitive performance with state-of-the-art VAE-based models in tasks such as text-to-image generation and image captioning.

**Strengths:**

1. JetFormer processes text and images within a single autoregressive transformer, removing the need for modality-specific encoders and supporting seamless multimodal generation tasks.
2. JetFormer’s fully end-to-end training from raw data enables it to potentially learn task-specific representations, enhancing adaptability without relying on pre-trained embeddings.

**Weaknesses:**

1. The claim that pre-trained modality-specific encoders (e.g., VQ-VAE) limit performance due to task-agnostic design lacks sufficient quantitative or qualitative evidence to substantiate this limitation convincingly.
2. While end-to-end training in JetFormer may offer the advantage of learning task-specific latent representations, this approach could also introduce challenges in terms of training stability and computational cost. The paper does not provide a comparative analysis between end-to-end and two-stage methods (e.g., VAE-based embeddings followed by generative model training), leaving it unclear whether the benefits of end-to-end learning outweigh the potential costs and stability issues.
3. JetFormer’s approach of combining text and image modalities into a unified embedding space lacks clear justification. Without evidence of improved multi-task performance or multimodal synergy, it is unclear if a shared embedding space is necessary or beneficial for this model’s architecture.

**Questions:**

(suggestion) To validate the advantages of JetFormer’s end-to-end learned image embeddings, additional experiments are needed to compare its performance against models using well-established VAE-based embeddings. Such comparative analysis would clarify whether JetFormer’s learned embeddings offer benefits over pre-trained embeddings in terms of image generation quality and task-specific adaptability.

---

> ### Author Response · Authors · 2024-11-23
>
> We thank the reviewer for their thoughtful comments, which we address in the following.
>
> 1. **Limitations of modality-specific pretrained encoders:** The limitations of pre-trained, modality-specific encoders in the context of multimodal LLMs are well documented and quantified in the literature, see for example [1, 2, 3]. Specifically, [1, Fig. 2] illustrates a clear tradeoff for between T2I and I2T model performance when varying the either number of T2I or I2T training examples, while keeping the number of I2T and T2I training examples fixed, respectively. The solution proposed in [1] to avoid this tradeoff is to decouple the representations for understanding and generation. We added these references to the corresponding claim in the introduction (in fact [1, 3] are already cited in our initial submission).
> Furthermore, we provide a qualitative examples in the appendix of the original submission: Fig. 5 compares reconstructions of the trained normalizing flow in JetFormer (replacing the factored-out dimensions with zeros) (column c) with those obtained with a (VQ-)VAE as used in VQGAN/MaskGIT/GIVT (column f). It can be seen that details such as text are much better preserved by JetFormer’s normalizing flow.
>
> 2. **Training stability and comparative analysis:** We did not observe training instabilities or similar issues. Indeed, end-to-end training of the normalizing flow and the transformer decoder is fully differentiable and does not change the training objective (data likelihood) but merely adds a log-determinant term (see Sec. 3.1). \
> For the suggested comparative analysis we train a VAE following the design of VQGAN/MaskGIT producing a sequence of 256 128-dimensional tokens like the normalizing flow in JetFormer. We remove the GAN and perceptual losses to ensure a fair comparison with JetFormer (which solely maximizes the data likelihood, without relying on GAN or perceptual losses). We then train the JetFormer-B decoder-only model on this VAE representation, following the JetFormer-B T&I training recipe. The results in the table below show that JetFormer outperforms the VAE baseline by a solid margin (in particular in T2I generation).
>
> |                               | COCO-FID30k | INet 0-shot | COCO Cap | VQAv2 |
> |-------------------------------|:-------------:|:-------------:|:----------:|:-------:|
> | VAE + Transformer (2 stages)  | 95.4        | 37.6        | 103.9    | 64.0    |
> | VAE + Transformer (2 stages, w/ noise curriculum)      | 87.6        | 40.0          | 106.7    | 64.9  |
> | JetFormer (end-to-end)        | 30.6        | 42.7        | 113.9    | 65.4  |
>
>
> 3. **Lack of clear justification:** We already showed in item 2. above that end-to-end training is clearly superior to using pretrained embeddings for both T2I and I2T tasks in the absence of specialized losses for image generation. \
> In a broader context, JetFormer is the first model which enables multimodal generative pretraining on raw data via next-token prediction, by solely optimizing the data likelihood, akin to pretraining ubiquitous in LLMs. Besides avoiding specialization of the image representation (and its limitations), this also enables architecture-agnostic comparison of validation likelihoods (and hence hill-climbing) and scaling laws (as those do not depend on a specific visual tokenizer used). Furthermore, compared to the best prior pixel likelihood-based image generation methods such as Image Transformer (Parmar et al. 2018) and iGPT (Chen et al. 2020), JetFormer’s image generation capabilities are a large step forward in terms of resolution and visual quality, and this at computational requirements comparable with VQGAN. \
> As a first of its kind approach, expecting JetFofmer to beat VQ-based approaches based on 100s or 1000s of incremental improvements from prior publications is an extremely high bar. We believe fully end-to-end paradigms like JetFormer could come to full fruition in the future if the current scaling trends in compute and unification across modalities continues.
>
>
> [1] Pan, Kaihang, et al. "Auto-Encoding Morph-Tokens for Multimodal LLM." ICML 2024 \
> [2] Xu, Yifan, et al. "Libra: Building Decoupled Vision System on Large Language Models." ICML 2024 \
> [3] Dong, Runpei, et al. "DreamLLM: Synergistic Multimodal Comprehension and Creation." ICLR 2024

---

> > ### Comment · Reviewer_9Pqq · 2024-11-27
> > **Response to rebuttal**
> >
> > I thank the authors for their efforts in addressing the concerns. While the experimental results are still not entirely convincing in supporting the use of unified embeddings, I appreciate the pioneering nature of this work and have decided to raise my score to 6.

---

### Public Comment · ~Michael_Tschannen1 · 2025-05-19

Code is available at https://github.com/google-research/big_vision.

---

### Meta-Review · Area_Chair_QiMp · 2024-12-24

**Metareview:**

The paper introduces JetFormer, an autoregressive generative model that handles raw image and text data in an end-to-end fashion without relying on pre-trained components like VQ-VAEs. The model employs a normalizing flow for encoding images into soft tokens, which are then processed by a decoder-only transformer together with the text. The paper shows that JetFormer achieves reasonable performance in text-to-image generation and image understanding, with much simpler architecture and training pipelines than existing multimodal models.

This paper opens up a new possibility in end-to-end training of multimodal autoregressive models. The proposed framework would be also inspiring for modalities beyond vision and language where pre-trained tokenizers are unavailable. It represents an important initial step in exploring this direction, and I believe it holds significant potential to inspire further advancements in the field. Although the performance is not state-of-the-art yet, it does not outweigh the significance of this work. Therefore, I recommend acceptance of the paper.

**Additional Comments On Reviewer Discussion:**

The major concern raised by the reviewers is on the performance of the model, particularly in image generation on ImageNet 256, where the reported results are not comparable to state-of-the-art benchmarks. While this issue was not fully addressed during the rebuttal, the reported performance remains relatively reasonable for a first attempt. On the other hand, the reviewers recognize the novelty and significance of the proposed end-to-end multimodal training approach. In my view, the novelty and potential impact of this work outweigh the performance limitations, making it a valuable contribution to the field.

---

### Decision · Program_Chairs · 2025-01-22

Accept (Poster)